# *IL17RB* and *IL17REL* Expression Are Associated with Improved Prognosis in HPV-Infected Head and Neck Squamous Cell Carcinomas

**DOI:** 10.3390/pathogens12040572

**Published:** 2023-04-07

**Authors:** Yuhan Sun, Md. Abdullah Al Kamran Khan, Stefano Mangiola, Alexander David Barrow

**Affiliations:** 1Department of Microbiology and Immunology, The Peter Doherty Institute for Infection and Immunity, The University of Melbourne, Melbourne 3000, Australia; 2Division of Bioinformatics, Walter and Eliza Hall Institute, Parkville 3052, Australia; 3Department of Medical Biology, The University of Melbourne, Melbourne 3010, Australia

**Keywords:** head and neck cancer, HPV infection, secretome, *IL17RB*, *IL17RE*-like proteins

## Abstract

Changes in the cellular secretome are implicated in virus infection, malignancy, and anti-tumor immunity. We analyzed the association between transcriptional signatures (TS) from 24 different immune and stromal cell types on the prognosis of HPV-infected and HPV-free head and neck squamous carcinoma (HNSCC) patients from The Cancer Genome Atlas (TCGA) cohort. We found that HPV-positive HNSCC patients have tumors with elevated immune cell TS and improved prognosis, which was specifically associated with an increased tumor abundance of memory B and activated natural killer (NK) cell TS, compared to HPV-free HNSCC patients. HPV-infected patients upregulated many transcripts encoding secreted factors, such as growth factors, hormones, chemokines and cytokines, and their cognate receptors. Analysis of secretome transcripts and cognate receptors revealed that tumor expression of *IL17RB* and *IL17REL* are associated with a higher viral load and memory B and activated NK cell TS, as well as improved prognosis in HPV-infected HNSCC patients. The transcriptional parameters that we describe may be optimized to improve prognosis and risk stratification in the clinic and provide insights into gene and cellular targets that may potentially enhance anti-tumor immunity mediated by NK cells and memory B cells in HPV-infected HNSCC patients.

## 1. Introduction

Head and neck squamous cell carcinomas (HNSCC) are the most common type of malignancies that arise in the head and neck. Over 90% of head and neck cancers are HNSCC that develop from the mucosal epithelium in the oronasal cavity, pharynx, and larynx. In the past two decades, accumulating studies have revealed the link between prior infection with oncogenic human papillomavirus (HPV) strains and tumors that develop in the oropharynx. An estimated 30,000 oropharyngeal cancers were caused by HPV infection worldwide each year, and HPV has been detected in ~25% of all HNSCC patients [1]. Despite the progress made in treatments, the prognosis of HNSCC patients has not improved significantly and HNSCC patients frequently suffer complications, such as local relapses and metastases [2]. Modulating the secretome in HNSCC and by HPV infection has recently been implicated in tumor progression, cancer cell invasion and metastasis [3].

The interleukin 17 (IL17) cytokine family contains six structurally related members from *IL17A* to *IL17F* (encoded by *IL17A*–*IL17F*). Five relevant IL17 receptor (IL17R) proteins have also been described as *IL17RA* through to *IL17RE* (encoded by *IL17RA*–*IL17RE*, respectively). The IL17 family has pivotal roles in inflammation, autoimmune disease, and cancer. *IL17A* (IL17) is the prototypic member produced by RAR-related orphan receptor gamma t (RORγt)-expressing cells and predominantly expressed by TH17 cells [4]. *IL17A* binding and activation of the IL17RA/IL17RC heterodimer signaling complex induces the expression of cytokines and chemokines such as tumor necrosis factor (TNF), CXC-chemokine ligand 1 (CXCL1), CXCL2 and CXCL5, from macrophages and stromal cells that are critical for defense against extracellular pathogens [5,6]. In contrast, dysregulation of *IL17A* expression is implicated in inflammatory disorders, such as psoriasis, ankylosing spondylitis and psoriatic arthritis [7,8]. The role of *IL17A* in cancer also remains controversial [9]. *IL17A* usually indirectly shapes immune suppression and helps tumor cell proliferation during the early stages of cancer by upregulating phosphorylated ERK1/2, angiogenesis and self-renewal [10,11,12], but also shows clinical relevance in anti-tumor immunity since IL17-secreting cells also co-secrete anti-tumor factors, such as IFN-γ and TNF [9,13,14].

While the knowledge of *IL17A* has been well established, the immunological roles of other IL17 family members are less well known. Human IL17B is a monomer that only shares 21.3% homology with human *IL17A* in the amino acid sequence [15]. In contrast to IL17A, IL17B expression has been detected in naïve, memory, germinal center B cells and chondrocytes as well as neurons, but not in activated T helper (TH) cells [16,17,18]. IL17B binds to the *IL17RB* homodimer, which is expressed in the epithelial cells of various organs [19]. The expression of *IL17RB* has also been found in TH cells and innate lymphocytes [20,21]. *IL17RB* can combine with *IL17RA* to form a heterodimeric complex that recognizes IL17E and prevents the binding of IL17B to *IL17RB*. In inflammatory diseases, the IL17B/IL17RB pathway is protective and might restrict pro-inflammatory by the IL17E/IL17RA-IL17RB signaling complex [22].

Several reports have shown the association between the IL17B/IL17RB signaling pathway and tumor development in breast, gastric, lung, pancreas, prostate, brain and blood cancers, however the precise mechanisms involved remain unclear [23,24,25,26]. Like IL17B, IL17C is frequently detected in non-immune cells. The binding of IL17C and IL17RA/IL17RE complex plays an important role in inflammation by regulating the innate immune functions of epithelial cells [27,28,29]. However, *IL17RE* has also been found to be expressed on TH17 cells in addition to stromal cells and *IL17RE* signaling can amplify T_H_17 cell responses in autoimmune disease [30]. In intestinal malignancies, IL17C binding to *IL17RE* stimulated TH17 cells to produce pre-tumorigenic cytokines and deficiency of *IL17RE* dramatically decreased intestinal tumor growth [31]. Collectively, these studies implicate the IL17 family of cytokines and receptors in tumor-promoting inflammation.

Following the discovery of the IL17 cytokine family, a gene encoding a novel IL17RE-like protein (IL17REL) was defined by similarity searches of amino acid sequences [32,33]. Rare variants of *IL17REL* with minor allele frequency (MAF) of less than 0.01 have been associated with inflammatory bowel disease (IBD), ulcerative colitis (UC) and gout. However, the role of *IL17REL* in malignancies remains uncertain [33,34,35].

Accumulating evidence has revealed the role of the IL17 cytokine family members in regulating the migration and functions of germinal center (GC)-derived B cells [16,36,37,38]. The GC is a transitory structure consisting of proliferating B cells in primary follicles of secondary lymphoid organs. The construction of the GC is divided into light and dark zones, formed by B cells following different developmental and functional patterns. The centrocytes (CC) in the light zone can differentiate further into memory B cells and plasma cells, and will go through further selection based on the affinity of the antibodies they produce [39,40,41]. It has been demonstrated that *IL17A* and IL17B were both expressed in the GC microenvironment, dedicated to B cell recruitment and antibody production [16,36,37].

NK cells are innate lymphocytes with cytotoxic and cytokine-secreting functions that comprise approximately 10–15% of total lymphocytes in human peripheral blood. They are known to initiate potent anti-tumor immunity regulated by “missing-self” and “induced-self” recognition [42,43,44]. NK cell activation is regulated by the balance of signaling from an array of activating and inhibitory surface receptors that bind to extracellular ligands [44,45]. In addition to conventional ligands anchored on the target cell surface, secreted molecules may also play roles in NK cell anti-tumor immunity. For example, one study revealed the secreted platelet-derived growth factor (PDGF)-DD as a ligand for the activating NK cell receptor, NKp44. PDGF-DD binding to NKp44 induces NK cell activation and secretion of IFN-γ and TNF and the transcription of mRNAs encoding proinflammatory chemokines, such as CCL3, CCL4, XCL1 and XCL2 [46]. The secretion of IFN-γ and TNF induces tumor cell growth arrest as well as the expression of ligands for the activating NK cell receptors, DNAM-1 and CRTAM, which initiate NK cell tumor surveillance [47,48,49,50]. In contrast to a well established pro-tumorigenic role in angiogenesis, high vascular endothelial growth factor (VEGF) expression in tumor-associated macrophages (TAM)/stroma was found to be associated with better prognosis in primary colon carcinoma, [51]. These data show that the tumor secretome may influence anti-tumor immunity and cancer patient prognosis.

Tumor immune infiltration is largely linked to the survival of a variety of HPV-related cancers. It has been demonstrated that high volumes of tumor-infiltrating lymphocytes (TILs) is associated with good prognosis in HPV-related cancer [52,53]. In addition to CD8^+^ T cells, which are a well-established lymphocyte subset associated with improved clinical outcomes [54,55], B cell markers have also been reported to improve the overall survival (OS) of HNSCC patients [56] and HPV-specific B cell phenotypes have been defined in the HNSCC tumor microenvironment [57]. NK cells have also been shown to play a protective role in HPV-associated cervical cancer immunotherapies [58]. However, the infiltration of specific immune cell states and their prognostic values remain unclear.

In this study, we set out to test the prognostic values of HPV infection and the expression of transcripts encoding components of the cellular secretome, such as growth factors, cytokines, chemokines, hormones, and their cognate receptors in HNSCC patients. We have applied a 24-cell-type TS and provided an overview of the TIL profiles in HPV-negative and -positive HNSCC patients as well as their prognostic associations. We find that a high abundance of activated NK cell or memory B cell TS are associated with good prognosis in HPV-infected HNSC patient tumors. Moreover, we also find that two genes encoding IL17 receptor family members, *IL17RB* and *IL17REL*, are associated with improved prognosis and may modulate the anti-tumor activity of memory B, NK cells, and T cell subsets in HPV-infected HNSCC patients.

## 2. Materials and Methods

### 2.1. TCGA Data Collection and Viral Load Estimation

We first collected RNA-seq data and clinical results for 500 tumor biopsies in the GDC Data Portal [59]. According to the study published by Cantalupo’s group, HPV viral alignments were detected by either RNA-seq or DNA-seq in 88 TCGA-HNSCC tumor biopsies, which were considered HPV-infected patients, whilst 412 tumor samples were deemed uninfected without any aligned HPV viral sequences. We took the maximum aligned viral sequences from HNSCC patients as defined by the Cantalupo group [60] for measuring HPV viral load.

### 2.2. Generation of Transcriptional Signatures and Deconvolution

The processes of the generation and benchmark of 24 cell type transcriptional signatures were described in our former publications [61,62]. Briefly, we first collected 592 highly curated (i.e., for which identity was confirmed in the literature), non-redundant biological replicates for 24 different immune and stromal cell types. The expected value and variability of gene transcription abundance for each cell type was then estimated by a Bayesian statistical model known as CellSig (github: stemangiola/cellsig), based on a negative binomial data distribution [63]. Afterwards, the transcriptional markers were selected by the pairwise comparison of each cell type within cell type categories along the cell differentiation hierarchy, which together formed the transcriptional signature (TS) matrix, as described [61,62].

### 2.3. RNA-seq Data Manipulation and Differential Expression Analysis

Raw read counts from TCGA-HNSCC RNA-seq data were scaled with the trimmed mean of M values (TMM) method [64] to compensate for differences in sequencing depth. Differential expression analysis was performed through edgeR quasi-likelihood dispersion driven by Rstudio between HPV-infected and HPV-free groups according to the publication of Cantalupo’s group [60] without other covariates (Appendix A). The threshold log fold change (logFC) ≥ 1.5 or log fold change ≤ −1.5 and false discovery rate (FDR) ≤ 0.05 defined the differentially significant genes. The significant genes were further classified and labeled by growth factor, cytokine, chemokine, hormone and their relevant receptors downloaded from the KEGG database [65].

Single-cell RNA-seq (scRNA-seq) analysis has also been performed to understand the cell type expressing *IL17RB* and *IL17REL*. Four open-access HNSCC single-cell RNA sequencing datasets from GEO: GSE139324 [66], GSE103322 [67], GSE164190 [68] and GSE173647. All analysis was performed based on the Seurat [69,70,71,72] package under the R environment.

### 2.4. Statistical Analysis

We estimated the cell type relative fractions for each biological replicate with our reference RNA-seq-derived transcriptional signature and the RNA-seq data from TCGA-HNSCC based on CIBERSORT [73]. Then, Kaplan–Meier (KM) survival curves were estimated from the median split CIBERSORT-inferred cell type fractions through the R framework tidybulk [74], with progression-free survival information as the measure of outcomes for HNSCC patients. The quantity percent survival versus time-to-event statistics was produced by the log-rank (Mantel–Cox) test [75]. The statistics of KM curves were adjusted using the Benjamini–Hochberg (BH) procedure. Further, the correlation analysis was performed by Pearson’s correlation test with default adjusted *p*-values.

Data analysis and visualization were performed using the R environment in RStudio. Packages include tidybulk [74], tidyHeatmap [76], survminer [77], survival [78], foreach [79], org.Hs.eg.db [80], cowplot [81], ggsci [82], GGally [83], gridExtra [84], reshape [85], Hmisc [86], and scales [87].

## 3. Results

### 3.1. HPV-Infected HNSCC Patients Have Increased Immune Cell TS Expression in HNSCC Tumors and More Favorable Survival

A previous study has shown that HNSCC patients have significantly lower rates of metastases with HPV infections, suggesting that viral infection may enhance cancer immune surveillance and influence HNSCC patient prognosis [88]. To understand whether HPV-infected HNSCC patients have improved prognosis, we compared the survival of HPV-positive and HPV-free HNSCC patients from 500 HNSCC tumors with both OS and last contact day information in the TCGA-HNSCC cohort (TCGA-HNSCC) (Figure 1). We found that HPV-positive patients (n = 88) had significantly better clinical outcomes than HPV-free patients (n = 412, Figure 1A).

Previous studies have reported that viral infection may enhance anti-tumor immunity in cancer patients by activating type-I interferon signaling [89,90,91]. To estimate the effect of viral load on tumor immune infiltration, we compared tumor expression of immune cell transcriptional signatures (TS) [61,62] between 88 HPV-infected and 412 HPV-free tumors in 500 primary HNSCC tumor biopsies (Figure 1B). Interestingly, the expression of immune cell TS and the gene encoding the common leukocyte antigen, CD45 (*PTPRC*), were significantly upregulated in HPV-infected compared to HPV-free HNSCC tumors, respectively (Figure 1B,C). Next, we asked whether the abundance of immune cell TS and *PTPRC* correlate with HPV viral load in HNSCC patient samples. The expression of immune TS and *PTPRC* was positively correlated with HPV viral load in HNSCC tumors (Figure 1D,E). Moreover, HPV-infected HNSCC patients with high viral load and high tumor abundance of immune TS had significantly improved prognosis compared to the rest of the cohort (Figure 1F). Our results show that HPV-positive HNSCC patient tumors have more abundant immune cell TS, which is associated with improved prognosis compared to HPV-negative HNSCC patients. We conclude that HPV-positive tumors are more immunogenic than HPV-free tumors, which is associated with an improved clinical outcome in HNSCC patients.

### 3.2. Increased Tumor Abundance of Memory B and NK Cell TS Are Associated with Improved Prognosis in HPV-Infected HNSCC Patients

To further understand the immune cell types associated with improved prognosis in HPV-infected HNSCC tumors, we compared the expression of TS from 24 different immune and stromal cell types in HPV-positive and HPV-free patients (Figure 2A–C). We found that TS of monocytes, immature dendritic cells (iDC), memory B cells, resting NK cells (ReNK) and activated NK cells (aNK) were more abundant in HPV-infected patient tumors (Figure 2B,C). Memory B cells and NK cells play important roles in anti-tumor immunity [46,92,93,94,95,96,97]. We wanted to understand the prognostic values of the cell-type-specific TS in HNSCC patients. Intriguingly, HNSCC patients with higher abundance of either memory B or aNK TS had improved prognosis compared to HPV-free patients (Figure 2D), but not for other immune cell TS (Appendix A). NK cells are known to be regulated by the missing-self recognition [44]. To understand whether the downregulation of MHC-I molecules at the transcriptional level was associated with the aNK TS, we compared the expression and combined survival of six MHC-I molecules (encoded by *HLA-A*, *-B*, *-C*, *-E*, *-F*, and *-G*) with the aNK TS (Appendix A). Interestingly, we found *HLA-E* was significantly downregulated, whilst its relevant receptors CD94 (encoded by *KLRD1*) and NKG2A (encoded by *KLRC1*) were significantly upregulated in HPV-infected patients (Appendix A). The combined KM survival plot showed that HPV-infected patients with both low MHC-I and high aNK TS expression had improved prognosis for *HLA-A*, *-C* and *-F* (Appendix A). Similar trends were also observed for *HLA-B*, *-E* and *-G*. These results indicate that HPV infection may boost anti-tumor immunity by increasing tumor infiltration of memory B cells and activated NK cells in HNSCC patients.

### 3.3. Secretome Genes and Cognate Receptors Are Differentially Expressed in HPV-Infected HNSCC Patients

Given that growth factor pathways may contribute to anti-tumor immunity [46,62], we were interested in discovering other potential biomarkers in the secretomes of HPV-infected patients that may contribute to anti-tumor immunity. Among all the genes that differentially expressed (logFC ≥ 1.5 or logFC ≤ −1.5 and FDR ≤ 0.05) in HPV-infected HNSCC patients (Appendix A), growth factor or relevant receptor genes *NGF*, *CSF2*, *EREG*, *IL1F10*, *FGF19*, and *FGFBP2* were downregulated, whereas *CSPG5*, *IGBPL1*, *IGFALS*, and *BMP3* were upregulated in HPV-infected tumors (Figure 3A). As for cytokine, chemokine, and relevant receptor genes, *NGF*, *IL1RL1*, *CSF2*, *IL1F10*, *CXCL5*, and *IFNK* were downregulated, and *IL17RB*, *IL17REL*, *CCL25*, and *BMP3* were upregulated with HPV infection (Figure 3B). The hormone and related receptor genes *GAL*, *RXFP1*, and *CGB5* were downregulated, whilst *EHNO*, *NPPG* and *TG* were upregulated in HPV-infected HNSCC patients (Figure 3C). In addition, we tested the correlation between the upregulated secretome genes and immune- and stromal-cell-type TS and HPV infection status to provide insights into possible pathways or cell responses in HPV-positive and HPV-negative HNSCC patients (Figure 3D, Appendix A). Apart from *BMP3,* which was positively correlated with the fibroblast TS in HPV-infected HNSCC patients, most upregulated secretome genes analyzed were positively associated with immune cells TS in HPV-infected patients compared to uninfected HNSCC patients (Figure 3D). Notable positive correlations for growth factors and receptors existed between: *IGFBPL1* (encoding insulin-like growth factor-binding protein 1)/Helper T, γδ T cell (GD) T and monocyte TS; *BMP3* (encoding bone morphogenetic protein 3)/fibroblast (Fibro) TS; *IGFALS* (encoding acid labile subunit)/T helper cell and GD T TS; *CSPG5* (encoding chondroitin sulfate proteoglycan 5)/naive CD8 T and M2 macrophage TS; and cytokines/chemokines and receptors: *IL17REL*/Treg, T Helper, aNK, monocyte, mast cell and memory B TS; *CCL25* (encoding chemokine C-C motif ligand 25)/Treg and GD T, CD8 Tem, IL2NK, monocyte, Mast cells and memory B cells TS; *IL17RB*/Treg, GD T, naïve CD8 T, CD4 Tem, aNK, monocyte, M2 macrophage and memory B TS; and hormones and receptors: *ENHO* (encoding adropin)/naïve CD8 T. CD8 Tcm, CD4 Tem, aNK, Macro M2 TS; *NPPC* (encoding natriuretic peptide precursor C)/Treg, GD T cell, aNK, monocytes and memory B TS; and *TG* (encoding thyroglobulin)/GD T cell, CD4 Tem, monocyte, mast cell and memory B TS in HPV-infected HNSCC tumors (Figure 3D). Overall, the upregulation of secretome genes was correlated with immune cell TS rather than stromal cell TS in HPV-infected HNSCC patients compared to HPV-free HNSCC patients, suggesting that secretome-encoded transcripts may play a role in anti-tumor immunity and improved prognosis in HPV-infected HNSCC patients.

### 3.4. Expression of IL17RB and IL17REL Are Associated with Improved Prognosis in HPV-Positive HNSCC Patients

We wanted to understand the relationship between the expression of genes encoding growth factors (Figure 4A, Appendix A), cytokines and chemokines (Figure 4B, Appendix A), hormones (Figure 4C, Appendix A) and their cognate receptors on the prognosis of HNSCC patients and viral load. In HPV-free HNSCC patients, the expression of all selected secretome genes did not influence prognosis except for *CCL25* (Figure 4B). In contrast, higher expression of *IL17REL* was associated with improved prognosis in HPV-positive HNSCC patients, while higher expression of *CSPG5*, *IL17RB*, *CCL25*, *NPPC* and *TG* trended towards improved prognosis and *ENHO* trended towards poor prognosis in HPV-positive HNSCC patients (Figure 4B). Moreover, consistent with upregulation in HPV infection, expression of all secretome genes was positively correlated with HPV viral load (Figure 4A–C).

We next performed a combined survival analysis using all curated secretome genes upregulated in HPV-infected HNSCC tumors (Appendix A) with HPV viral load in HNSCC patients (Figure 5 and Appendix A). Of the secretome genes analyzed, only higher expression of *IL17RB* and *IL17REL* was associated with improved prognosis in HPV-infected HNSCC patients, which was more marked in those patients with higher HPV viral loads (Figure 5 and Appendix A) compared to HNSCC patients with lower HPV loads. Finally, higher expression of *IL17A*, which encodes the known anti-tumoral cytokine *IL17A* [9,13,14], was associated with improved prognosis (Appendix A). These results show that higher expression of *IL17RB* and *IL17REL* are associated with higher viral load and improved prognosis of HPV-positive HNSCC patients.

### 3.5. High Expression of IL17RB and IL17REL and TIL Subset TS Are Associated with Improved Prognosis in HPV-Infected HNSCC Patients

Since higher expression of *IL17RB*, *IL17REL*, and the memory B and aNK TS are associated with improved prognosis in HPV-infected HNSCC patients, we aimed to understand whether these IL17 receptor family genes may cooperate with certain immune or stromal cell types for improved prognosis in HNSCC patients by carrying out a combined survival analysis based on tumor expression of either *IL17RB* or *IL7REL* and our immune and stromal cell TS in HPV-infected HNSCC patients (Figure 6). HNSCC patients with higher tumor expression of *IL17RB* combined with high expression of either the CD4 Tem, memory B cell, or aNK TS had improved prognosis compared to patients with lower expression of *IL17RB* and high expression of either the CD4 Tem, memory B cell, or aNK TS (Figure 6, top panel). Moreover, HNSCC patients with higher tumor expression of *IL17REL* combined with high expression of either the CD8 Tcm, memory B, M1 macrophages, or Helper T cell TS also had improved prognosis compared to those HNSCC patients with lower tumor expression of *IL17RB* and high expression of either CD8 Tcm, memory B, M1 macrophages or Helper T cell TS (Figure 6, bottom panel). These trends were not observed for any other immune or stromal cell TS in either HPV-positive or HPV-negative HNSCC patients (Appendix A).

In addition to our investigation, we analyzed four publicly available HNSCC scRNA-seq datasets, namely GSE139324 [66], GSE103322 [67], GSE164190 [68] and GSE173647, but were unable to detect significant read counts for *IL17RB* or *IL17REL* (Appendix A). Our results suggest that the expression of *IL17RB* may influence the anti-tumor functions of CD4 Tem, memory B, and activated NK cells, and *IL17REL* may influence the anti-tumor functions of CD8Tcm, memory B cells, M1 macrophages, and Helper T cells in HPV-positive HNSCC tumors.

## 4. Discussion

Head and neck squamous cell carcinomas (HNSCC) are among the most common cancers worldwide, with over 870,000 new cases and 440,000 deaths in 2020 [98]. Smoking, chewing tobacco, alcohol and HPV infection are the main risk factors for HNSCC. The prognosis of HNSCC is known to be implicated by a group of host and tumor characteristics, including pathological differentiation grading, performance status, and tumor node metastasis (TNM) staging. HPV infection has been described to alleviate distant metastases and is associated with improved survival of HNSCC patients [88,99,100,101,102,103]. In this study, we aimed to determine the impact of transcripts encoding secreted factors, such as growth factors, hormones, chemokines and cytokines, and cognate receptors, and immune cell TS on the prognosis of HPV-infected and uninfected HNSCC patients to provide insights for biomarkers and future therapies that may target the secretome in HNSCC.

The immune response has been implicated in the development of HNSCC and HPV infection [104,105]. We hypothesized that the improved prognosis of HPV-infected HNSCC patients resulted from enhanced immune responses induced by viral infection at the tumor site. Tumor abundance of TS representing immune cells and the gene (*PTPRC*) encoding the common leukocyte antigen, CD45, were associated with improved prognosis in HPV-infected HNSCC patients compared to uninfected patients and positively correlated with HPV viral loads, suggesting HPV infection may enhance anti-tumor immunity in HNSCC patients. Analysis of the tumor abundance of 24 different cell-type-specific TS [61,62] showed that activated NK cells and memory B cells are associated with the improved prognosis of HPV-infected HNSCC patients. It has been reported that the gene expression profiles and tumor microenvironment (TME) in HPV-infected tumors are different from HPV-free malignancies [106]. Interestingly, we have found both resting NK cell and activated NK cell TS to be upregulated in HPV-infected patients. Since the reduction of MHC class I gene expression was identified as a hallmark of HPV-infected biopsies [107], these results may suggest increased recruitment and activation of NK cells in HNSCC tumors after HPV infection. Indeed, we found significant downregulation of transcripts encoding HLA-E in HPV-infected compared to HPV-free patients, and combined survival analysis showed that HPV-infected patients with low expression of *HLA-A*, *-C*, and *-F* and high tumor abundance of activated NK cells were associated with improved prognosis (Appendix A). Memory B cells that initiate rapid immune responses were enriched in HNSCC tumor sites [57,108,109]. It has been reported that the genital HPV vaccine may help prevent cancers that develop in the oral cavity [110] and induce long-term protection by developing high-affinity memory B cells [111,112], indicating the importance of memory B cells in HPV-related cancers.

Interestingly, our study revealed no increase in tumor abundance of T cell subsets, in HPV-positive HNSCC patients, contrary to previous reports suggesting that favorable prognosis in HNSCC patients was mediated by CD8^+^ GZMA^+^ PRF1^+^ T cells [113]. Moreover, consistent with T cell exhaustion due to prolonged antigen stimulation in viral infections and cancer [114], HPV-infected HNSCC tumors showed significant upregulation of T cell exhaustion markers in the TCGA-HNSCC dataset [115]. However, our T cell transcriptional signatures were based on normal T cell subsets, which may have limited our ability to detect exhausted T cells enriched in HPV-positive HNSCC tumor sites. Proteomic approaches to identify candidate secreted protein biomarkers of prognostic significance in HNSCC have been reported, but are limited by small patient cohorts [3,116,117]. In our study, we chose a computational approach focused on understanding the association of transcripts encoding secreted molecules and their cognate receptors and the immune system on the prognosis of HPV-infected patients from the TCGA-HNSCC cohort comprising 500 patient samples. Taking this approach, we have clarified the prognostic values of two genes encoding proteins from the IL17 receptor family, *IL17RB* and *IL17REL*, in HNSCC that were significantly upregulated in HPV-infected compared to uninfected HNSCC patients. The IL17 signaling pathway has previously been found to be associated with poor prognosis in cancers, whilst the role of *IL17REL* in cancers remains unknown. Interestingly, expression of the gene for the current known ligand for *IL17RB*, *IL17B*, did not show any correlation with immune cell TS or survival (Appendix A).

*IL17REL* encodes an IL17RE-like protein originally discovered using a similarity search for novel members of the IL17 receptor family. All IL17 receptor family sequences share the same conservative intracellular signaling motif named SEF/IL17R (SEFIR) [32]. However, IL17RE-like proteins lack the SEFIR motif but instead resembles the extracellular domain of *IL17RE* [32]. Though *IL17REL* was found to be highly associated with gout and ulcerative colitis in genome-wide association studies (GWAS) [33,35], the expression and function of *IL17REL* in different cell types remain largely unknown, especially in human cancers. In contrast, *IL17RE* is expressed on both epithelial and TH17 cells and recognizes IL17C as ligand. IL17C binding to *IL17RE* induces IL17 production from TH17 cells, which can enhance host innate immunity [118]. We conducted single-cell RNA sequencing (scRNA-seq) analysis on four publicly available HNSCC datasets to identify the cells expressing *IL17RB* or *IL17REL*. However, in contrast to bulk RNA-seq data, scRNA-seq data had a lower resolution for detecting *IL17RB* or *IL17REL* (Appendix A). Thus, it awaits to be seen whether IL17RE-like has similar immune functions to *IL17RE* and how these might influence prognosis in HPV-infected HNSCC patients.

We have also found a positive correlation between *IL17RB*, *IL17REL* and memory B cell and aNK TS expression (Figure 3D). It has been previously studied that in germinal centers, CXCR4^+^ or CXCR5^+^ GC B cells were induced to migrate to CXCL12 or CXCL13 through the *IL17RB* signaling pathway [16]. Strikingly, CXCR4 has also been defined as a critical molecule in NK cell trafficking, and reduced CXCR4 expression significantly impaired NK cell migration in vitro [119,120,121]. It may be possible that increased expression of *IL17RB* and *IL17REL* may participate in the recruitment of memory B and activated NK cells in HPV-infected HNSCC patients.

To summarize, our study set out to uncover a possible role for genes encoding secretome factors or their receptors and their association with immune and stromal cell TS and HPV infection on the prognosis of HNSCC patients. We have uncovered a novel association between a higher tumor expression of *IL17RB* with CD4 Tem, memory B, and activated NK TS and *IL17REL* with CD8 Tcm, memory B, M1 macrophages and T Helper cell TS and the improved prognosis of HPV-infected HNSCC patients. Our results have important consequences for anti-tumor immunity and reveal potential new biomarkers or targets for immunotherapy in HPV-infected HNSCC patients.

## Figures and Tables

**Figure 1 pathogens-12-00572-f001:**
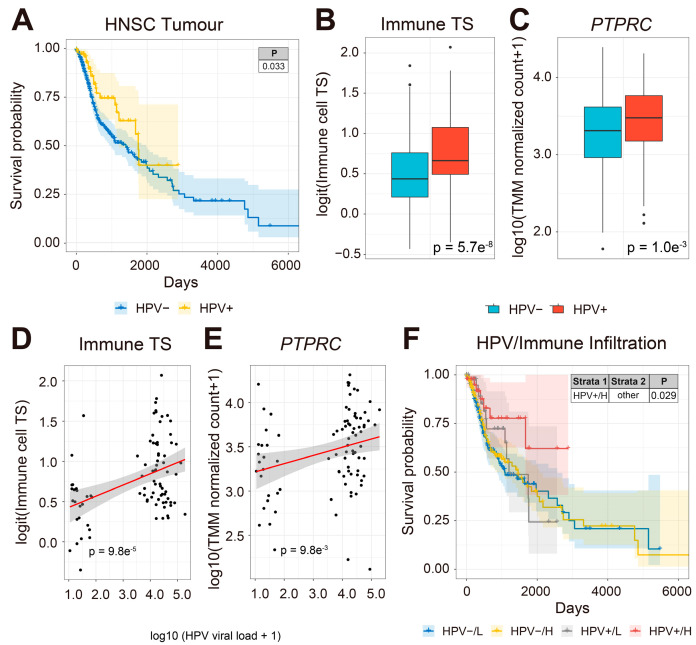
HPV infection and immune cell transcriptional signature are associated with improved prognosis in HNSCC patients. (**A**) KM survival curve comparing HPV-infected (yellow line) and HPV-free (blue line) HNSCC patients (y-axis, survival probability; x-axis, days). HPV-infected patients have improved survival compared to HPV-free HNSCC patients. (**B**) Box plots comparing HNSCC tumor expression of immune cell TS and (**C**) *PTPRC* (CD45). (**D**) Scatter plots of the correlation (red line) between HPV viral load (x-axis) and the expression of immune cell TS (y-axis) and **(E)** *PTPRC* (y-axis) in HPV-infected HNSCC patients. The abundance of immune cell TS and *PTPRC* is positively correlated with viral load in HPV-infected HNSCC tumors. (**F**) Combined KM survival plot (y-axis, survival probability; x-axis, days) of HPV-infection status and tumor expression of total immune cell TS. The KM curve represents HNSCC patient survival plotted in all four combinations for each stratum (HPV%#x2212;/L, HPV-%#x2212;/H, HPV+/L, and HPV+/H) with total tumor expression of immune cell TS split by the median into L (low) and H (high) patient groups. HPV-infected HNSCC patients with high expression of immune cell TS have significantly better clinical outcomes compared to all other groups.

**Figure 2 pathogens-12-00572-f002:**
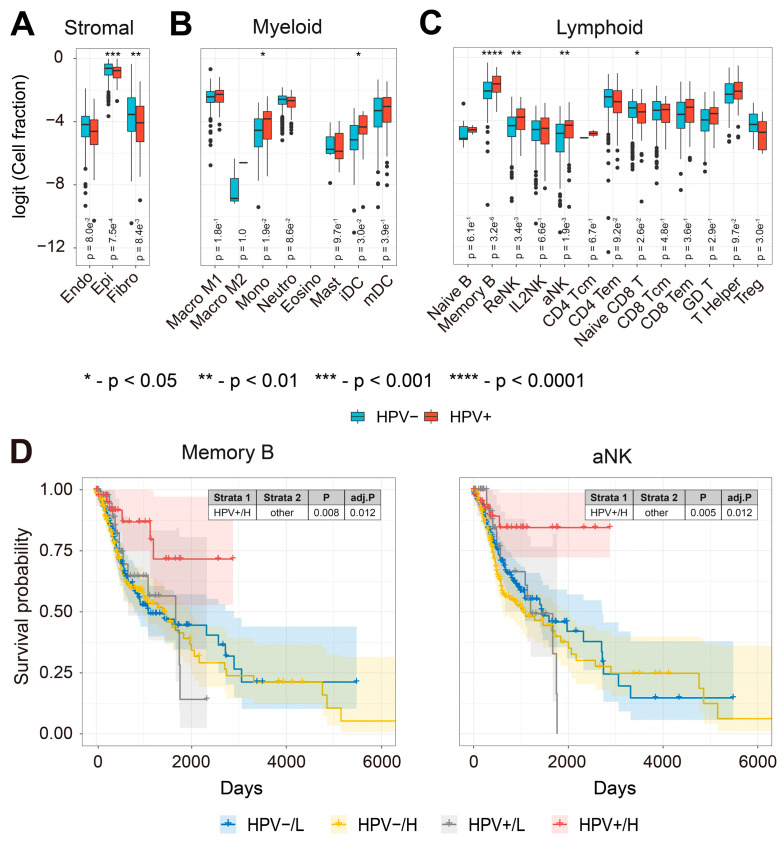
TS of Memory B and activated NK cells are associated with improved prognosis in HPV-positive HNSCC patients. (**A**) Box plots comparing the abundance of stromal cell TS (endothelial cells, epithelial cells, and fibroblasts); (**B**) myeloid cell TS (M1 and M2 macrophages, monocytes, neutrophils, eosinophils, mast cells, immature and mature dendritic cells); (**C**) lymphoid cell TS (naïve B cells, memory B cells, resting NK cells (ReNK), IL2-primed NK cells (IL2NK), activated NK cells (aNK), central (Tcm) and effector memory (Tem) CD4^+^ T cells, naïve CD8 T cells, Tcm and Tem CD8^+^ T cells, γδ T cells, Helper T cells and regulatory T cells (Treg) (* refers to *p*-value < 0.05, ** refers to *p*-value < 0.01, *** refers to *p*-value < 0.001, and **** refers to *p*-value < 0.0001); (**D**) KM survival curves (y-axis, survival probability; x-axis, days) constructed for combinations of HPV infection and memory B cell and aNK TS expression in HNSCC patient tumors. Each cell’s TS expression was split by the median into L and H groups. HNSCC patients with both HPV infection and high expression of either memory B cell TS or aNK TS had significantly improved prognosis compared to other groups.

**Figure 3 pathogens-12-00572-f003:**
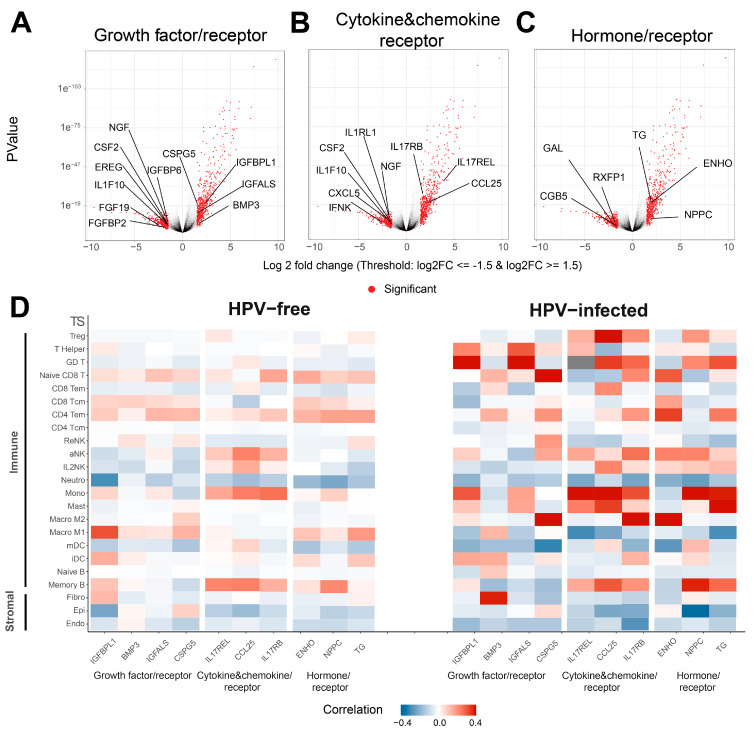
**Figure 3**. Identification of differentially expressed secretome genes in HPV-infected HNSCC tumors. (**A**) Volcano plots showing significantly downregulated or upregulated growth factor, (**B**) cytokine and chemokine, and (**C**) hormone and relevant receptor genes in HPV-infected HNSCC tumors. (**D**) Correlation heatmap of significantly upregulated secretome genes and all cell TS in HPV-free and -infected HNSCC patients.

**Figure 4 pathogens-12-00572-f004:**
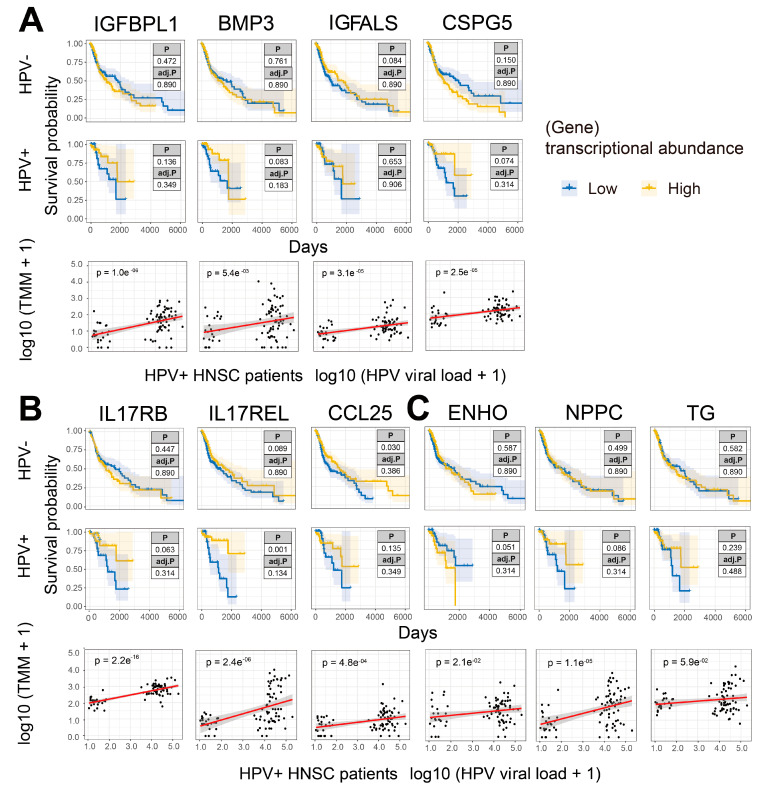
Upregulated secretome (and receptor) genes are associated with improved HNSCC prognosis and are positively correlated with viral load. (**A**) KM curves (y-axis, survival probability; x-axis, days) constructed for growth factor, (**B**) cytokine and chemokine, (**C**) hormone and relevant receptor genes that were upregulated in HPV-infected HNSCC patients and having a significant correlation with viral loads in HPV-infected patients. Each gene expression was split by the median into L and H groups. The scatter plots were constructed to correlate the above genes and HPV viral loads in HPV-infected patients. High tumor expression of *IL17REL* was significantly associated with improved prognosis, while another IL17 family member *IL17RB* showed the same trend but without significance. The expression of both *IL17REL* and *IL17RB* was significantly positively correlated with HPV viral loads in infected HNSCC patients.

**Figure 5 pathogens-12-00572-f005:**
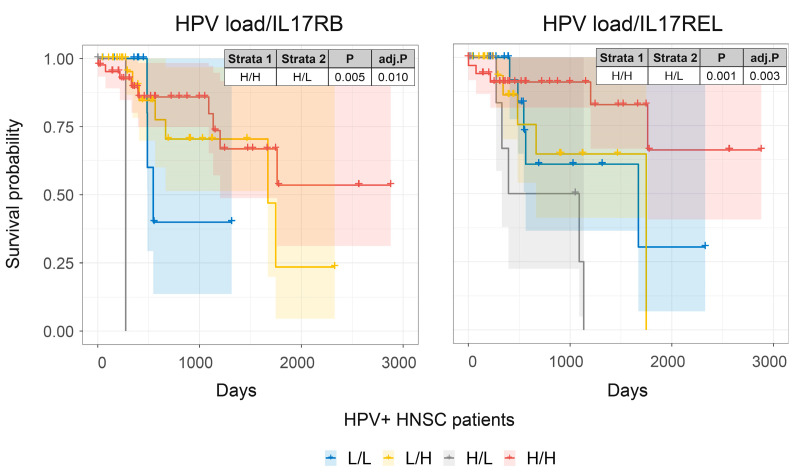
The association of IL17 family receptors and survival in HPV-infected HNSCC patients. Combined HNSCC patient survival analysis stratified for HPV viral loads and IL17 family members, *IL17RB* and *IL17REL*, respectively. KM curves (y-axis, survival probability; x-axis, days) display HPV-infected HSNC patient survival plotted in all four combinations for each stratum (L/L, L/H, H/L, and H/H, both L and H groups were split by the median viral load or gene expression). For patients with higher HPV viral loads, high expression of either *IL17RB* or *IL17REL* resulted in enhanced prognosis.

**Figure 6 pathogens-12-00572-f006:**
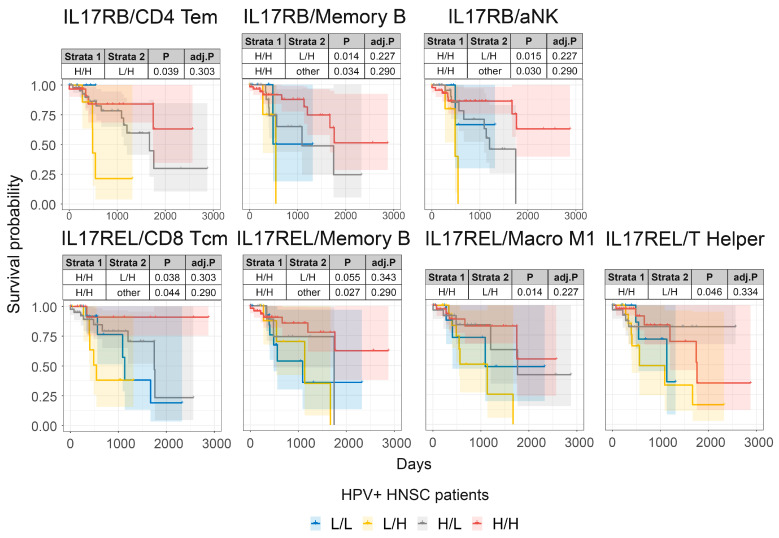
The combined survival analysis of *IL17RB*, *IL17REL* and immune cell TS in HPV-infected HNSCC patients. Combined HNSCC patient survival analysis stratified for *IL17RB* and *IL17REL* and TIL TS expression in HPV-infected HNSCC tumors. KM curves (y-axis, survival probability; x-axis, days) display HPV-infected HSNC patient survival plotted in all four combinations for each stratum (L/L, L/H, H/L, and H/H, both L and H groups were split by the median viral load or gene expression). For patients with high TIL TS expression, higher *IL17RB* or *IL17REL* expression was associated with an improved prognosis.

## Data Availability

The codes for all figures (including Appendix A) are available at https://github.com/RAGG3D/HPV-TCGA-HNSCC (accessed on 16 March 2023).

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
