# Peer review of "IL17RB and IL17REL Expression Are Associated with Improved Prognosis in HPV-Infected Head and Neck Squamous Cell Carcinomas"

_pathogens, 2023, doi:10.3390/pathogens12040572_

Round 1

Reviewer 1 Report

The here presented work by Sun et al. is a nice analysis of RNA-seq data of human HNSCC samples with special emphasis on host immunity and signalling. First of all I'd appreciate if the authors use the commonly used abbreviation HNSCC for head and neck squamous cell carcinoma.

The here presented results are somewhat expectable since it is well known that HPV+ HNSCC have better prognosis with increased host immunity being one of the major reasons for this observation. It would have been nice to see whether costimulatory molecules on immune cells or MHC molecules on cells are up or downregulated in HPV+ vs HPV- HNSCC to gain further insight in tumour microenvironment and show proof of concept for this approach since MHC I downregulation should be visible.

Interestingly T cells or monocytes/DC for instance were not increased in HPV+ samples. Unfortunately the authors do not discuss this somewhat striking observation but solely discuss B and NK cells which they have shown to be overrepresented in their data. Especially since the observed expression of IL-17RE is associated with T cells.

The discussion of IL-17REL which seems according to Sun et al. relevant for favorable prognosis in HPV+ HNSCC is somewhat frustrating since only one sentence mentions an unknown role. What does this molecule do? What signalling cascades can be induced by IL-17REL. Does this molecule at all have relevance in immune modulation, activation, migration, cancer immunity,... and if yes by which mode of action? I do believe it to be a relevant result since a corresponding receptor is also increased in their RNA-seq data and not an artifact. Yet the authors need to provide conclusive information why their findings are relevant for increased survival in HPV+ HNSCC patients. 

Overall this study has a lot of potential and I encourage to resubmit a revised version to Pathogens.

Author Response

Responses to the referees’ comments

# Reviewer 1

The here presented work by Sun et al. is a nice analysis of RNA-seq data of human HNSCC samples with special emphasis on host immunity and signalling. First of all I'd appreciate if the authors use the commonly used abbreviation HNSCC for head and neck squamous cell carcinoma.

Our response:

We have edited the text accordingly.

The here presented results are somewhat expectable since it is well known that HPV+ HNSCC have better prognosis with increased host immunity being one of the major reasons for this observation.

Our response:

We appreciate that the association of HPV infection and HNSCC prognosis has already been described in Wu et al. Frontiers in Oncology 2021, Gameiro et al. Oncoimmunology 2018, Yu et al. Frontiers in Oncology 2022. However, our transcriptional signature represents an improvement over previous works by achieving higher resolution of cell types and significantly increasing the scale of our reference signature datasets (Sun et al. 2021). Our cell TS also estimates the proportions of cell states (e.g. resting NK cell and activated NK cell). With this approach, we made new observations that activated NK cells and memory B cells may be important as well as their association with IL17REL and IL17RB expression and improved prognosis which is not a widespread observation in the field. We have revised the manuscript to make this clearer in the Introduction section.

Page 3, line 123

In this study, we set out to test the prognostic values of HPV infection and the expression of transcripts encoding components of the cellular secretome, such as growth factors, cytokines, chemokines, hormones and their cognate receptors in HNSCC patients. We have applied a 24 cell type TS and provided an overview of the TIL profiles in HPV negative and positive HNSCC patients as well as their prognostic associations. We find that a high tumor abundance of activated NK cell or memory B cell TS are associated with good prognosis in HPV infected patients. Moreover, we find two genes encoding IL-17 receptor family members, IL17RB and IL17REL, to be associated with improved prognosis in HPV infected HNSCC patients.”

It would have been nice to see whether costimulatory molecules on immune cells or MHC molecules on cells are up or downregulated in HPV+ vs HPV- HNSCC to gain further insight in tumour microenvironment and show proof of concept for this approach since MHC I downregulation should be visible.

Our response:

We thank the Reviewer #1 for this feedback. Following on from our observation that activated NK cells may be important in the HNSC tumour microenvironment, we estimated the association between the transcription abundance of six MHC-I genes (HLA-A, -B, -C, -E, -F, and -G) with the HPV status (Supplementary Figure 3).

The following results were added to the manuscript in the Results 3.2 subsection:

Page 7, line 247

Interestingly, we found HLA-E was significantly downregulated whilst its relevant receptors CD94 (encoded by KLRD1) and NKG2A (encoded by KLRC1) were significantly up-regulated in HPV infected patients (Supplementary Figure 3A). The combined KM survival plot showed that HPV infected patients with both low MHC-I and high aNK TS expression had improved prognosis for HLA-A, -C and -F (Supplementary Figure 3B). Similar trends were also observed for HLA-B, -E and -G.”

Interestingly T cells or monocytes/DC for instance were not increased in HPV+ samples. Unfortunately the authors do not discuss this somewhat striking observation but solely discuss B and NK cells which they have shown to be overrepresented in their data. Especially since the observed expression of IL-17RE is associated with T cells.

Our response:

We agree with Reviewer #1 that this is an important aspect. Our estimates (Figure 2) indeed show the enrichment of monocytes and dendritic cells in the HPV+ samples.

We have discussed this association in the Discussion subsection.

Page 13, line 406

Interestingly, our study revealed no increase in tumour abundance of T cell subsets, in HPV-positive HNSCC patients, contrary to previous reports suggesting that favorable prognosis in HNSCC patients was mediated by CD8+ GZMA+ PRF1+ T cells [113]. Moreover, consistent with T cell exhaustion due to prolonged antigen stimulation in viral infections and cancer [114], HPV-infected HNSCC tumors showed significant upregulation of T cell exhaustion markers in the TCGA-HNSCC dataset [115]. However, our T cell transcriptional signatures were based on normal T cell subsets, which may have limited our ability to detect exhausted T cells enriched in HPV-positive HNSCC tumor sites.”

The discussion of IL-17REL which seems according to Sun et al. relevant for favorable prognosis in HPV+ HNSCC is somewhat frustrating since only one sentence mentions an unknown role. What does this molecule do? What signalling cascades can be induced by IL-17REL. Does this molecule at all have relevance in immune modulation, activation, migration, cancer immunity,... and if yes by which mode of action? I do believe it to be a relevant result since a corresponding receptor is also increased in their RNA-seq data and not an artifact. Yet the authors need to provide conclusive information why their findings are relevant for increased survival in HPV+ HNSCC patients.

Our response:

We agree that defining the context of the IL-17REL protein would benefit the article. However, there is very limited information on this gene which has mostly described in only four studies Franke et al. Nature Genetics 2010, Wu et al. Immunogenetics 2011, Sasaki et al. Inflammatory Bowel Diseases 2016, and Dong et al. Protein and Cell 2017. We have added the following paragraph to the Discussion section, to cover as much of the currently available knowledge on IL-17REL as possible.

Page 13, line 431

Though IL-17REL was found to be highly associated with gout and UC in genome-wide association study (GWAS) [33,35], the expression and function of IL-17REL in different cell types remain largely unknown, especially in human cancers.

Reviewer 2 Report

In this study, Barrow group tried to explain why HPV-positive HNSCC has a better prognosis and one reason could be increases in immunogenic abundance impacted by HPV, represented by high expression of IL17RB and IL17REL in the tumor. The whole story was built up based on statistical data analysis without further biological experiment verification. The manuscript should be taken into consideration for publication with major changes, as suggested as follows.

1.     If the authors aimed to figure out the immune cell behaviors in the tumor, why don’t use single cell sequencing data, which I believe is also available to the public?

2.     Line 103: Why introduce PDGF-DD? Any relation with the later context? It is not necessary to describe so much if the authors did not study this term in their result.

3.     Line 110: Lack the explanation of DNAM-1 and CRTAM.

4.     Line 181: What exactly genes are included in this immune cell transcriptional signature? What is the importance of CD45?

5.     Line 215: I can’t tell M2, mast cells and aNK cells were more abundant in HPV-positive patients. M2 and mast cells come with a “ns” label, which makes me even confused.

6.     Section 3.3 is very loosely related to the other sections. Basically the authors described changes of bunch of growth factors, cytokines, or hormones in the tumor. In line 260 the authors stated that secretome genes were correlated with immune cell TS, and how? At least the authors should provide some experiment to verify one of those targets that could contribute to immune infiltration.     

7.     In the figure 4, if CCL25 was considered as a potent target, then why IGFBPL1 is not in HPV+  samples?

Reviewer 3 Report

In the manuscript entitled: “Tumor expression of IL17RB and IL17REL are associated with improved prognosis in HPV-infected head and neck squamous cell carcinomas”, the authors analyze the transcriptional signatures (TS) including immune and stromal cells in HPV-positive and HPV-negative Head and Neck Squamous Carcinomas (HNSC) of patients from The Cancer Genome Atlas (TCGA) RNA-seq database, aiming to define the impact of transcripts of secreted factors (growth factors, hormones, chemokines, cytokines, cognate receptors), and transcriptional signatures of immune infiltrating cells in the clinical outcome. They found that upregulation of IL17RB and IL17REL transcripts is associated with transcriptional signatures defining higher HPV load, and high abundance of memory B cells and NK cells. The authors conclude that HNSC HPV-positive tumors are more immunogenic than HPV-negative tumors, which is associated with an improved clinical outcome in HNSC patients.

I consider that this is an interesting and well-designed work with relevant information in the field.

First, the authors confirmed that patients with HPV-positive HNSC tumors have better overall survival than HPV-negative patients. Further, expression profile analyses showed that immune cell transcriptional signature (TS) as well as CD45 were upregulated in HNSC HPV positive tumors compared with HPV negative tumors and interestingly those profiles had a positive correlation with HPV viral load. Moreover, HNSC patients with tumors with high viral load and high immune transcriptional signature had an improved clinical outcome. When dissecting the cell types associated with the immune TS, they found that specifically patients with HNSC HPV-positive tumors with higher abundance of memory B or NK TS had improved prognosis compared to patients with HPV-negative tumors or with low abundance of those immune cells.  When analyzing secretome genes they found that the upregulation of IL17RB and IL17REL was associated with high viral load and with improved prognosis in patients with HPV-positive HNSC.

I have few comments:

-The title:  change “..are associated…”, for “..is associated..”

- as such, there are grammatical errors that should be revised.

- The introduction is too heavy and specifically related to IL17RB and IL1REL, it could be improved including information about immune infiltration and clinical outcome in HPV-related cancers and why the intention to carry out this type of analysis.In the manuscript entitled: “Tumor expression of IL17RB and IL17REL are associated with improved prognosis in HPV-infected head and neck squamous cell carcinomas”, the authors analyze the transcriptional signatures (TS) including immune and stromal cells in HPV-positive and HPV-negative Head and Neck Squamous Carcinomas (HNSC) of patients from The Cancer Genome Atlas (TCGA) RNA-seq database, aiming to define the impact of transcripts of secreted factors (growth factors, hormones, chemokines, cytokines, cognate receptors), and transcriptional signatures of immune infiltrating cells in the clinical outcome. They found that upregulation of IL17RB and IL17REL transcripts is associated with transcriptional signatures defining higher HPV load, and high abundance of memory B cells and NK cells. The authors conclude that HNSC HPV-positive tumors are more immunogenic than HPV-negative tumors, which is associated with an improved clinical outcome in HNSC patients.

I consider that this is an interesting and well-designed work with relevant information in the field.

First, the authors confirmed that patients with HPV-positive HNSC tumors have better overall survival than HPV-negative patients. Further, expression profile analyses showed that immune cell transcriptional signature (TS) as well as CD45 were upregulated in HNSC HPV positive tumors compared with HPV negative tumors and interestingly those profiles had a positive correlation with HPV viral load. Moreover, HNSC patients with tumors with high viral load and high immune transcriptional signature had an improved clinical outcome. When dissecting the cell types associated with the immune TS, they found that specifically patients with HNSC HPV-positive tumors with higher abundance of memory B or NK TS had improved prognosis compared to patients with HPV-negative tumors or with low abundance of those immune cells.  When analyzing secretome genes they found that the upregulation of IL17RB and IL17REL was associated with high viral load and with improved prognosis in patients with HPV-positive HNSC.

I have few comments:

-The title:  change “..are associated…”, for “..is associated..”

- as such, there are grammatical errors that should be revised.

- The introduction is too heavy and specifically related to IL17RB and IL1REL, it could be improved including information about immune infiltration and clinical outcome in HPV-related cancers and why the intention to carry out this type of analysis.

Author Response

Reviewer 3

In the manuscript entitled: “Tumor expression of IL17RB and IL17REL are associated with improved prognosis in HPV-infected head and neck squamous cell carcinomas”, the authors analyze the transcriptional signatures (TS) including immune and stromal cells in HPV-positive and HPV-negative Head and Neck Squamous Carcinomas (HNSC) of patients from The Cancer Genome Atlas (TCGA) RNA-seq database, aiming to define the impact of transcripts of secreted factors (growth factors, hormones, chemokines, cytokines, cognate receptors), and transcriptional signatures of immune infiltrating cells in the clinical outcome. They found that upregulation of IL17RB and IL17REL transcripts is associated with transcriptional signatures defining higher HPV load, and high abundance of memory B cells and NK cells. The authors conclude that HNSC HPV-positive tumors are more immunogenic than HPV-negative tumors, which is associated with an improved clinical outcome in HNSC patients.

I consider that this is an interesting and well-designed work with relevant information in the field.

First, the authors confirmed that patients with HPV-positive HNSC tumors have better overall survival than HPV-negative patients. Further, expression profile analyses showed that immune cell transcriptional signature (TS) as well as CD45 were upregulated in HNSC HPV positive tumors compared with HPV negative tumors and interestingly those profiles had a positive correlation with HPV viral load. Moreover, HNSC patients with tumors with high viral load and high immune transcriptional signature had an improved clinical outcome. When dissecting the cell types associated with the immune TS, they found that specifically patients with HNSC HPV-positive tumors with higher abundance of memory B or NK TS had improved prognosis compared to patients with HPV-negative tumors or with low abundance of those immune cells.  When analyzing secretome genes they found that the upregulation of IL17RB and IL17REL was associated with high viral load and with improved prognosis in patients with HPV-positive HNSC.

I have few comments:

-The title:  change “..are associated…”, for “..is associated..”

Our response:

We thank the Reviewer for pointing this out. We have amended.

- as such, there are grammatical errors that should be revised.

Our response:

We apologise for this, and we thoroughly revised the text rectifying any remaining grammatical errors and typos.

- The introduction is too heavy and specifically related to IL17RB and IL1REL, it could be improved including information about immune infiltration and clinical outcome in HPV-related cancers and why the intention to carry out this type of analysis.

Our response

We appreciate this feedback. We have added the following paragraph in the Introduction subsection.

Page 3, line 114

Tumor immune infiltration is largely linked to the survival of a variety of HPV-related cancers. It has been demonstrated that high volumes of tumor infiltrating lymphocytes (TILs) is associated with good prognosis in HPV-related cancer [52,53]. In addition to CD8+ T cells, which are a well-established lymphocyte subset associated with improved clinical outcomes [54,55], B cell markers have also been reported to improve the overall survival (OS) of HNSCC patients [56] and HPV-specific B cell phenotypes have been defined in the HNSCC tumour microenvironment [57]. NK cells were also identified to play essential roles in HPV-associated cervical cancer immunotherapies [58]. However, the infiltration of specific immune cell states and their prognostic values remain unclear.

Round 2

Reviewer 2 Report

no more questions.